# The Mechanosensitive Pkd2 Channel Modulates the Recruitment of Myosin II and Actin to the Cytokinetic Contractile Ring

**DOI:** 10.3390/jof10070455

**Published:** 2024-06-28

**Authors:** Pritha Chowdhury, Debatrayee Sinha, Abhishek Poddar, Madhurya Chetluru, Qian Chen

**Affiliations:** Department of Biological Sciences, The University of Toledo, 2801 Bancroft St, Toledo, OH 43606, USA; pchowdh3@rockets.utoledo.edu (P.C.); debatrayee.sinha@rockets.utoledo.edu (D.S.); madhurya.chetluru@rockets.utoledo.edu (M.C.)

**Keywords:** actin, cytokinesis, fission yeast, myosin, polycystin, Pkd2

## Abstract

Cytokinesis, the last step in cell division, separates daughter cells through mechanical force. This is often through the force produced by an actomyosin contractile ring. In fission yeast cells, the ring helps recruit a mechanosensitive ion channel, Pkd2, to the cleavage furrow, whose activation by membrane tension promotes calcium influx and daughter cell separation. However, it is unclear how the activities of Pkd2 may affect the actomyosin ring. Here, through both microscopic and genetic analyses of a hypomorphic *pkd2* mutant, we examined the potential role of this essential gene in assembling the contractile ring. The *pkd2-81KD* mutation significantly increased the counts of the type II myosin heavy chain Myo2 (+18%), its regulatory light chain Rlc1 (+37%) and actin (+100%) molecules in the ring, compared to the wild type. Consistent with a regulatory role of Pkd2 in the ring assembly, we identified a strong negative genetic interaction between *pkd2-81KD* and the temperature-sensitive mutant *myo2-E1*. The *pkd2-81KD myo2-E1* cells often failed to assemble a complete contractile ring. We conclude that Pkd2 modulates the recruitment of type II myosin and actin to the contractile ring, suggesting a novel calcium-dependent mechanism regulating the actin cytoskeletal structures during cytokinesis.

## 1. Introduction

Cytokinesis is the last stage of cell division when the daughter cells separate physically through mechanical force. In many eukaryotes, this process requires an actomyosin contractile ring, usually assembled at the equatorial division plane. Several key signaling pathways regulate the assembly and constriction of this contractile ring (for a review, see [1]). They include those mediated by the Polo kinase, the Central Spindlin complex, the Hippo/SIN (Septation Initiation Network)/MEN (Mitotic Exit Network) pathway and RhoA (for reviews, see [2,3,4,5]). These pathways in turn reorganize the actin cytoskeletal structures during cytokinesis to assemble the contractile ring, consisting of dozens of actin-binding proteins including formin, cofilin and myosin [6,7,8,9]. Among them, the type II non-muscle myosin (myosin II) plays a central role by driving both the assembly and disassembly of actin filaments in the contractile ring [9,10,11,12]. Myosin II is a motor whose activities depend on its heavy chain, the essential light chain and the regulatory light chain. In fission yeast, there are two myosin II heavy chains, Myo2 and Myp2, but only the former is essential for cytokinesis [13,14]. The essential light chain is Cdc4, while the regulatory light chain is Rlc1 [15,16,17]. It remains unknown how myosin II transits from its initial role of assembling the actin filaments in the ring to its later role of disassembling these filaments as the ring constricts.

We recently found that an essential ion channel, Pkd2, is required for cytokinesis of the unicellular model organism fission yeast *Schizosacchromyces pombe* [18,19]. Pkd2 belongs to the conserved family of polycystin channels. Most vertebrates possess two polycystins, PC-1 and PC-2 (for a review, see [20]). Mutations of human polycystins lead to a common genetic disorder, Autosomal Dominant Polycystic Kidney Disorder (ADPKD) [21,22]. However, the cellular function of polycystins, particularly in cell division [23], has remained largely unknown. Fission yeast Pkd2, reconstituted in vitro, allows the passage of Ca^2+^ when the membrane is stretched by mechanical force [24]. In vivo, Pkd2 is trafficked to the plasma membrane where its distribution is cell-cycle dependent [25]. Both its putative extracellular lipid-binding domain and the central nine-helix transmembrane domain are essential. In contrast, its C-terminal cytosolic tail, although dispensable, ensures its polarized distribution on the plasma membrane and selective internalization through the endocytic pathway [25]. During interphase, this channel mostly localizes at the cell tips where it promotes the tip extension of fission yeast cells [26]. During cytokinesis, Pkd2 is recruited to the cleavage furrow at the start of the furrow ingression, where it stays until the end of cell separation. This Ca^2+^-permeable channel, likely activated by the membrane stretching at the cleavage furrow, promotes the separationCa^2+^ spikes, a transient influx of Ca^2+^ during cytokinesis [24,27]. The activity of Pkd2 is essential for separating daughter cells. To our knowledge, Pkd2 is the only calcium channel required for cytokinesis in fission yeast.

Here, we tested the hypothesis that Pkd2 can regulate the assembly and constriction of the contractile ring during cytokinesis. Our hypothesis was largely based on the discovery of Pkd2 as a Ca^2+^ influx channel localized at the cleavage furrow. As an essential secondary messenger, Ca^2+^ could activate a large number of Ca^2+^-sensitive molecules including myosin regulatory light chains, calmodulin, calcineurin and Ca^2+^-dependent kinases. Many of them play a crucial role in regulating the actin cytoskeletal structures. In this study, we took advantage of a hypomorphic mutant of the essential *pkd2* gene. This mutation, *pkd2-81KD*, replaces the endogenous promoter with an repressible *81nmt1* promoter, reducing the expression of the *pkd2* gene by more than 70% under the suppressing condition [18]. One of the surprising phenotypes of this mutant is that it accelerates the cleavage furrow ingression, even though the mutant daughter cells often fail to separate [18]. Here, using quantitative fluorescence microscopy, we compared the molecule counts of the type II myosin heavy chain Myo2, the regulatory light chain Rlc1 and actin in the contractile ring of *pkd2-81KD* to the wild type. In addition to the microscopic analyses, we examined the genetic interaction between the *pkd2* mutants, including both *pkd2-81KD* and the new temperature-sensitive mutant, *pkd2-B42* [26], and the *myo2-E1* mutant. Overall, our results demonstrate that Pkd2 modulates the recruitment of both myosin II and actin to the contractile ring, suggesting that this force-activated channel may regulate the assembly of actin cytoskeletal structures during cytokinesis through Ca^2+^.

## 2. Materials and Methods

### 2.1. Cell Culture and Yeast Genetics

Yeast cell culture and genetics were carried out according to the standard protocols [28]. Unless specified, YE5S was used in all experiments. For genetic crosses, tetrads were dissected using Sporeplay+ dissection microscope (Singer, Reading, UK). Yeast strains used in this study are listed in Table 1. For ten-fold dilution series of yeast, overnight cultures were diluted and inoculated for additional 5 h before being spotted onto YE5S agar plates. The plates were incubated for 2–3 days in respective temperatures before being scanned using a photo scanner (Epson, Los Alamitos, CA, USA).

### 2.2. Microscopy

For microscopy, fission yeast cells were first inoculated in YE5S liquid media for two days at 25 °C before being harvested through centrifugation at 1500× *g* for 1 min during exponential phase at a density between 5 × 10^6^ cells/mL and 1.0 × 10^7^ cells/mL. The cells were resuspended in 50 µL of YE5S and 6 µL of the resuspended cells were spotted onto a gelatin pad (25% gelatin in YE5S) on a glass slide. The cells were sealed under a coverslip (#1.5) with VALAP (1:1:1 mixture of Vaseline, lanolin and paraffin) before imaging.

Live microscopy was carried out on a spinning disk confocal microscope equipped with EM-CCD camera. The Olympus IX71 microscope was equipped with objective lenses of 60× (NA = 1.40, oil) and 100× (NA = 1.40, oil), a motorized Piezo Z Top plane (Applied Scientific Instrumentation, Eugene, OR, USA) and a confocal spinning disk unit (Confocal Scanner Unit -X1, Yokogawa, Tokyo, Japan). The images were captured on an Ixon-897 EMCCD camera (Andor, Santa Barbara, CA, USA). Solid state lasers of 488 and 561 nm were used to excite green (GFP) and red fluorescence (tdTomato and mCherry) proteins, respectively. Unless specified, the cells were imaged with Z-series of 15 slices at a step size of 0.5 µm. Time-lapse microscopy was carried out in a dedicated dark room, maintained at ∼23 ± 2 °C. To minimize temperature variations among the experiments, we always imaged the wild type and mutant cells on either the same or consecutive days.

To stain the cell wall with calcofluor, exponentially growing cell culture was fixed with 4% paraformaldehyde, prepared from 16% fresh stock solution (Electron Microscopy Science, Hatfield, PA, USA). The fixed cells were washed with TEMK buffer (50 mM Tris-HCL pH 7.4, 1 mM MgCl_2_, 50 mM KCl, 1 mm EGTA, pH 8.0). The fixed cells were then stained with 1 µg/mL calcofluor, prepared with the stock solution (1 mg/mL, Sigma–Aldrich, Rockville, MD, USA) on a rocking platform for 10 min at room temperature in the dark. The stained cells were pelleted and resuspended in 50 µL TEMK, 6 µL of which was spotted on a glass slide directly and sealed under a coverslip (#1.5) with VALAP. The stained samples were imaged with an Olympus IX81 microscope equipped with 60× objective lens (NA = 1.42), a LED lamp for bright-field microscopy, and a mercury lamp for fluorescence microscopy. The micrographs were captured on an ORCA C-11440 digital camera (Hamamatsu, Japan), operated with the CellSens software (https://lifescience.evidentscientific.com.cn/en/software/cellsens/, accessed on 14 January 2024) (Olympus, Westborough, MA, USA).

### 2.3. Western Blots

For probing the expression level of GFP-Myo2, 50 mL of exponentially growing yeast cells were harvested by centrifugation at 1500× *g* for 5 min. We washed the cells with 1 mL of sterile water once. The pelleted cells were re-suspended with 300 µL of lysis buffer (50 mM Tris-HCl pH 7.5, 100 mM KCl, 3 mM MgCl_2_, 1 mM EDTA, 1 mM DTT, 0.1% Triton X-100 and protease inhibitors (Halt protease inhibitor cocktail; #1862209; Thermo Fisher, Waltham, MA, USA)). The cells were mixed with 300 mg of glass beads (0.5 mm) before being mechanically homogenized by a bead beater (BeadBug, Benchmark Scientific, Sayreville, NJ, USA) for five cycles of 1 min breaking interrupted by 1 min incubation on ice. The lysed cell was immediately mixed with pre-heated 5× sample buffer (250 mM Tris-HCL pH 6.8, 50% glycerol, 25% β-mercaptoethanol, 15% SDS, 0.025% Bromophenol Blue) before being heated for 10 min at 100 °C. After centrifugation at top speed for 1 min, the supernatant was collected for SDS-PAGE gel electrophoresis (Mini-PROTEIN TGX 10% precast, BioRad #4561033, Irvine, CA, USA). The gels were either stained with Coomassie blue or transferred to PVDF membrane (Amersham #10600023) for immunoblots. The membrane was blotted with anti-GFP monoclonal primary antibody (mouse monoclonal, Sigma-Roche, #11814460001) at 4 °C overnight, followed by horseradish peroxidase-conjugated Goat-anti-mouse second antibodies (1:10,000; BioRad, #1721011) for 1 h at room temperature. Blots were developed with chemi-fluorescent reagents (Pierce ECL Western Blotting Substrate, Thermo Scientific, #32209, MA, USA). The blots were quantified through ImageJ (https://imagej.net/ij/, accessed on 14 January 2024).

### 2.4. Image Analysis

We used ImageJ (NIH), its plugins and macros to process all the micrographs. For quantitative analysis, the time-lapse series were first corrected for X–Y drifting using the plugin StackReg [29] and photobleaching was corrected through the ImageJ plugin EMBLTools [30]. Average intensity projections of Z-series were used for quantification. The fluorescence intensity of GFP-Myo2 and Rlc1-tdTomato at the contractile ring was quantified by measuring the average fluorescence intensities in a 1 µm by 4 µm rectangle centered on the equatorial division plane. The measurements were corrected for the average background fluorescence, measured in a 0.2 µm by 4 µm rectangle adjoined to the equatorial division plane for both GFP-Myo2 and Rlc1-tdTomato. To measure the fluorescence intensities of GFP-Lifeact in a matured contractile ring, we manually segmented the ring at the start of the ring constriction based upon the fluorescence of Rlc1-mCherry. To determine when the ring starts to constrict, we analyzed the fluorescence kymograph constructed from the time-lapse series of a contractile ring. The total fluorescence of GFP-Lifeact in a ring equals the average fluorescence intensity of GFP-Lifeact in the ring, minus its cytoplasmic fluorescence, multiplied by the area of the ring (~1.3 µm^2^).

For quantification of immunoblots, the band intensities were measured by polygon tools of ImageJ and the measurements were corrected for the background fluorescence.

All the figures were prepared with Canvas X (ACD Systems) and the plots were prepared using Origin 2021 (OriginLab, Northampton, MA, USA).

## 3. Results

### 3.1. Pkd2 Modulates the Recruitment of Myo2, Rlc1, and Actin to the Contractile Ring

We first determined how Pkd2, as a Ca^2+^ influx channel localized at the cleavage furrow, regulates the localization of the myosin II heavy chain Myo2 to the contractile ring. To measure the relative number of Myo2 molecules in the *pkd2-81KD* mutant cells, we compared the fluorescence intensities of the N-terminally tagged GFP-Myo2, expressed from its endogenous locus, of the mutant to that of the wild type using quantitative microscopy [11] (Figure 1A). The intracellular concentration of GFP-Myo2, measured by either the fluorescence intensities (Figure 1B) or the immunoblots (Figure 1C), was unchanged in the mutant cells. During cytokinesis, similar to the wild type cells, the *pkd2* mutant cells recruited GFP-Myo2 to a broad band of cytokinesis nodes before condensing the nodes into a complete contractile ring (Figure 1D). However, the contractile ring of the *pkd2* mutant cells matured and constricted more quickly than that of the wild type cells (Figure 1D), as reported previously [18]. The fluorescence intensity of GFP-Myo2 in the contractile ring of the *pkd2* mutant cells increased more quickly and peaked sooner than the wild type (Figure 1E). As a result, the fluorescence intensity of GFP-Myo2 in a mature ring, referred to the ring just before its constriction, of the mutant increased by 18% (*p* < 0.01) (Figure 1F). Thus, Pkd2 modulates the recruitment of the myosin II heavy chain Myo2 to the contractile ring during cytokinesis.

Next, we examined whether Pkd2 helps recruit Rlc1, the regulatory light chain of myosin II [15], to the contractile ring. Rlc1 possesses four EF-hands which bind Ca^2+^. It is a potential target of the Ca^2+^ signaling pathway regulated by Pkd2. As above, to measure the relative number of Rlc1 molecules in the *pkd2* mutant cells, we compared the fluorescence intensity of Rlc1, tagged endogenously with the fluorescence protein tdTomato (Rlc1-tdTomato), in *pkd2-81KD* cells to that in the wild type cells (Figure 2A). Interestingly, we found a significant increase (26%, *p* < 0.001) in the intracellular fluorescence intensities of Rlc1-tdTomato of the *pkd2* mutant (Figure 2B). During cytokinesis, Rlc1-tdTomato was recruited to a broad band of nodes in the *pkd2* mutant cells before these nodes condensed into the contractile ring (Figure 2C). However, the *pkd2-81KD* mutant resulted in the fluorescence of Rlc1-tdTomato in the ring increasing much more quickly and peaking at least 10 min sooner (Figure 2D). Similar to Myo2, the average fluorescence intensity of Rlc1-tdTomato in a mature ring of the mutant was 37% higher than that of the wild type (*p* < 0.01) (Figure 2E). Therefore, in addition to the myosin II heavy chain, Pkd2 also modulates the recruitment of the myosin II regulatory light chain to the cytokinetic contractile ring.

Lastly, we determined whether Pkd2 has a role in the assembly of actin filaments in the cytokinetic contractile ring. We employed GFP-Lifeact as the fluorescence reporter [11] to measure the number of actin molecules in the ring. The endogenously expressed GFP-Lifeact was driven by a promoter of intermediate strength (Pcof1) to minimize the potential interferance with the actin cytoskeleton by this probe [31]. The *pkd2* mutation substantially altered all the actin cytoskeletal structures, including endocytic actin patches, actin cables and the contractile ring (Figure 3A). In particular, the number of actin patches increased significantly in the *pkd2-81KD* mutant cells, compared to the wild type. Compared to the wild type cells in which the actin patches concentrated at the cell tips, more patches of the mutant spread out in the middle portion of a cell (Figure 3A). Interestingly, the intracellular concentration of GFP-Lifeact in the mutant cells increased by almost 70% (*p* < 0.001) (Figure 3B). During cytokinesis, the assembly and disassembly of actin filaments, labeled by GFP-Lifeact, proceeded normally in the contractile ring of the mutant cells (Figure 3C). However, the fluorescence of GFP-Lifeact in the ring gradually increased before decreasing as the ring constricted. The total fluorescence of GFP-Lifeact in the mature contractile ring of *pkd2-81KD* cells doubled that of the wild type cells (*p* < 0.001) (Figure 3D). Based on these results, we concluded that in addition to myosin II, Pkd2 also modulates the assembly of actin filaments in the cytokinetic contractile ring.

### 3.2. Both pkd2 and myo2 Are Essential for the Contractile Ring Assembly in Cytokinesis

To further understand the relationship between *pkd2* and *myo2*, we examined their interactions through genetic studies of *pkd2* and *myo2* mutants. First, we confirmed the previously discovered negative genetic interaction between *pkd2-81KD* and a temperature-sensitive mutant *myo2-E1* (Figure 4A) [18]. Now, we similarly found a negative genetic interaction between the temperature-sensitive mutant *pkd2-B42* [26] and *myo2-E1* (Figure 4A). Even at the permissive temperature of 25 °C, the double mutant *pkd2-B42 myo2-E1* grew much more slowly than either *pkd2-B42* or *myo2-E1* (Figure 4A). Next, we compared the morphology of *pkd2-81KD, myo2-E1*, and *pkd2-81KD myo2-E1* cells with each other. Both *pkd2-81KD myo2-E1* and *pkd2-81KD* cells appeared to be rounder than the wild type cells (Figure 4B). Both of them also contained a small fraction of “deflated” cells, which shrank temporarily even in the rich YE medium (Figure 4B). However, the *pkd2-81KD myo2-E1* mutant stood out for its large fraction of lysed cells (11 ± 1.7%, average ± standard deviation) even at the permissive temperature of 25 °C, far more than either the *pkd2-81KD* or *myo2-E1* mutant (Figure 4B). A much higher fraction of the double mutant (74%) contained septum than either of the single mutants (Figure 4C,D). The septum of *pkd2-81KD myo2-E1* cells often appeared to be wavy, a rare phenotype among either of the single mutants (Figure 4C). Many more of the double mutant cells were multi-septated (21 ± 3%) than either *pkd2-81KD* or *myo2-E1* cells (Figure 4D). Based on these genetic analyses, we concluded that *pkd2* and *myo2* have a synergistic relationship in cytokinesis.

To determine how Pkd2 and Myo2 work synergistically in cytokinesis, we measured the contractile ring assembly and constriction in the *pkd2-81KD myo2-E1* cells. We used Rlc1-tdTomato as the marker for the contractile ring in these double mutant cells. We immediately noticed, even at the permissive temperature of 25 °C, that the fluorescent intensity of Rlc1-tdTomato in the contractile rings of *pkd2-81KD myo2-E1* cells appeared to be far lower than that of *myo2-E1* cells (Figure 5A). Quantitative measurement revealed that the intracellular fluorescence of Rlc1-tdTomato in the *pkd2-81KD myo2-E1* cells was 70% lower than that of *myo2-E1* (*p* < 0.001) (Figure 5B). Although the double mutant cells condensed the cytokinetic nodes into a contractile ring, most of these rings were incomplete with many gaps before the constriction ensued (n = 26) (Figure 5C). The fluorescence of Rlc1-tdTomato in the ring increased very slowly in the *pkd2-81KD myo2-E1* mutant cells, but it peaked at the same time as the wild type cells (Figure 5D). As a result, the average fluorescence of Rlc1-tdTomato in a mature ring decreased by 74% in the double mutant, compared to *myo2-E1* (*p* < 0.001) (Figure 5E). Surprisingly, these rings constricted at a rate comparable to those of either the wild type or *myo2-E1* cells at 25 °C, but much more slowly than *pkd2-81KD* cells (Figure 5F). We concluded that both the myosin II heavy chain Myo2 and Pkd2 work synergistically to recruit the myosin regulatory light chain Rlc1 to the contractile ring.

## 4. Discussion

In this study, we investigated a potential role of the mechanosensitive ion channel Pkd2 in the assembly and constriction of the actomyosin contractile ring during cytokinesis. We found that Pkd2 modulates the recruitment of the myosin II heavy chain, its regulatory light chain and actin to the contractile ring. The regulatory role of Pkd2 is synergistic with that of myosin II in promoting the assembly of a complete contractile ring.

The modulating role of Pkd2 in recruiting myosin II and actin to the contractile ring, although surprising, is consistent with the cytokinesis defects of the *pkd2* mutant cells. Our earlier works have demonstrated that the cleavage furrow ingressed almost 50% more quickly in the *pkd2* mutant cells compared to the wild type [18]. This is partially due to the reduced turgor pressure of the mutant cells [26]. Here, we found another factor that may have contributed to the faster ingression of the cleavage furrow in the *pkd2* mutant cells. The contractile ring of the mutant contains significantly more myosin II and actin molecules than the wild type. The myosin II heavy chain Myo2 increases in the ring of the *pkd2-81KD* mutant cells, but it is substantially less than the increase of the regulatory light chain Rlc1 and actin. Doubling the expression of the heavy chain Myo2 alone slows the contractile ring constriction in a previously published study [32]. In our *pkd2-81KD* mutant cells, the increase of three molecules, Myo2, regulatory light chain Rlc1 and actin, in the ring shall allow the ring to generate additional mechanical force during cytokinesis. Combining this increased compression force with the reduced resistance from the turgor pressure in the *pkd2* mutant cells, it is not surprising that the cleavage furrow ingresses much more quickly in *pkd2-81KD* mutant cells.

Due to the timing of the appearance of Pkd2 at the cleavage furrow, it appears unlikely that this ion channel directly recruits either myosin II or actin filaments to the contractile ring. Before cytokinesis, Pkd2 localizes at the cell tips. It only starts to appear at the cleavage furrow at the start of the furrow ingression when the contractile ring has been fully assembled. More likely, the modulating effect of Pkd2 in the recruitment of myosin II and actin to the ring is indirect and Ca^2+^-dependent. Although the role of Ca^2+^ in fission yeast cytokinesis remains unclear, this secondary messenger plays a critical role in cytokinesis of animal cells. In these cells, Ca^2+^-dependent myosin light chain kinase (MLCK) phosphorylates the myosin II regulatory light chain, triggering the contractile ring constriction [33]. However, such a mechanism of action by Ca^2+^ is not evolutionarily conserved. Unlike animal cells, fission yeast does not possess a homologue of MLCK. Instead, Rlc1 is phosphorylated by the P21-acivated kinases Pak1 and Pak2 during cytokinesis [34,35]. Although such modification of Rlc1 is not essential for cytokinesis under a normal growth condition [36], it does become necessary when the availability of actin is limited under a stress condition [35]. So far, there is no evidence that either Pak1 or Pak2 depend on Ca^2+^/calmodulin for their kinase activities. Thus, it remains in doubt whether these two kinases could be the downstream targets of Pkd2. Alternatively, the phosphorylation of Rlc1 may be regulated through a balance of phosphorylation by Pak1/2 and dephosphorylation by the calcium/calmodulin-activated phosphatase calcineurin Ppb1 [37,38]. Further work will be needed to illustrate how Pkd2 modulates the recruitment of myosin II through Ca^2+^.

In addition to the direct modulation of the recruitment of myosin II and actin molecules to the contractile ring, Pkd2 may regulate the transcription of these genes. This mechanism may offer an alternative explanation to why mutations of *pkd2* increase the intracellular concentration of Rlc1 and GFP-Lifeact. Specifically, the significant increase of intracellular concentration of GFP-Lifeact in the *pkd2* mutant cells suggests that the expression of this actin probe may be affected by the *pkd2* mutation. The elevated expression of this fluorescence actin probe in the *pkd2* mutant cells may partially contribute to the increased fluorescence of GFP-Lifeact in the contractile ring. Similarly, we have observed the elevated transcription of GFP in the temperature-sensitive *pkd2-B42* cells [26]. The molecular mechanism of such transcription regulation by Pkd2 remains unclear. One plausible scenario is that the cytoplasmic tail of Pkd2 may translocate into the nucleus to directly regulate transcription as its human homologues PC-1 does [39]. However, we have not observed even partial localization of Pkd2 in the nucleus so far. Alternatively, Pkd2 could indirectly regulate transcription through calcineurin. This Ca^2+^/calmodulin-dependent phosphatase Ppb1 promotes the nuclear translocation and activation of the transcription activator Prz1 [40]. Future work will determine the potential mechanism by which Pkd2 regulates gene transcription.

Considering the modulating effects of Pkd2 in the recruitment of Myo2 to the contractile ring, the negative genetic interaction between the *pkd2* and *myo2* mutants is surprising. Even more surprising is our discovery that the abundance of Rlc1 in the contractile ring decreased dramatically in the contractile ring of the *myo2 pkd2* double mutant cells. This is despite the fact that the *pkd2-81KD* mutation increases the recruitment of the regulatory light chain of Myosin II to the contractile ring. A possible explanation is that both Pkd2 and Myo2 promote the stability of the contractile ring. Pkd2 may stabilize the contractile ring through modulating the assembly of actin and myosin II. In contrast, the myosin heavy chain Myo2 promotes the contractile ring’s stability through both its motor and actin-binding activities, the latter of which is compromised in the *myo2-E1* mutant [9,41]. Our discovery suggests that both Pkd2 and Myo2 stabilize the contractile ring, although they do so through disparate mechanisms.

Overall, this study demonstrated that the force-sensitive ion channel Pkd2 regulates the assembly of the actomyosin contractile ring through modulating the recruitment of myosin II and actin to the ring. Our findings suggest that Ca^2+^ and Ca^2+^-binding proteins may play a critical role in reorganizing the actin cytoskeletal structures during fission yeast cytokinesis despite a lack of the canonical MLCK-dependent pathway.

## Figures and Tables

**Figure 1 jof-10-00455-f001:**
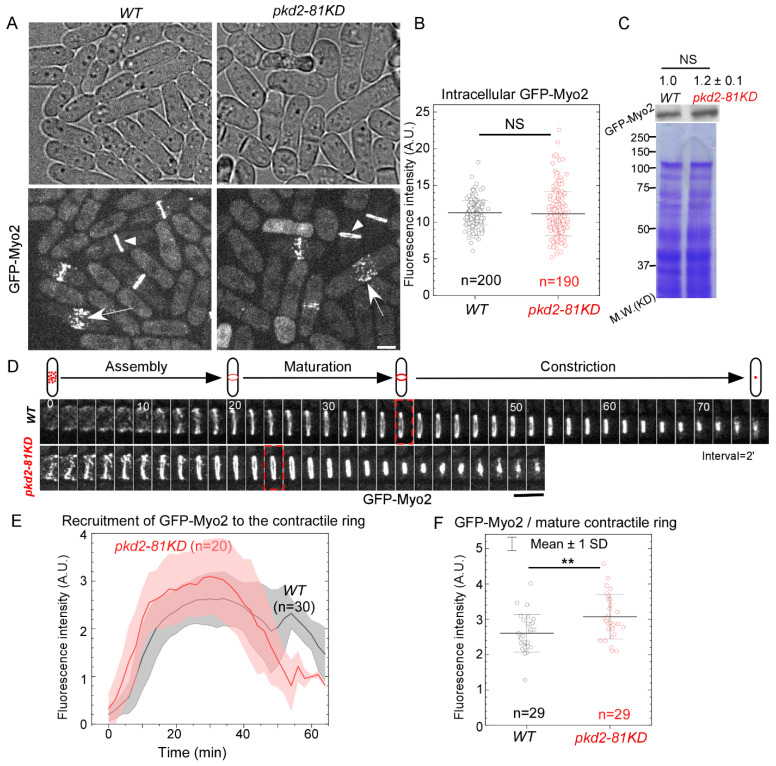
The *pkd2-81KD* mutant increased the recruitment of myosin II heavy chain Myo2 to the contractile ring. (**A**) Micrographs of wild type and *pkd2-81KD* cells expressing GFP-Myo2. White arrow: cytokinetic nodes; White arrowhead: the contractile ring. (**B**) Dot plot of the average intracellular fluorescence intensity of GFP-Myo2 in a cell. (**C**) Top: anti-GFP blot of the lysate from wild type and *pkd2-81KD* cells expressing GFP-Myo2. Number indicates the normalized intensity of GFP-Myo2 band ± standard deviations. Bottom: Coomassie blue stained SDS-PAGE gel of respective lysates. Representative data from three biological repeats are shown. (**D**) Top: a diagram of the contractile ring assembly, maturation and constriction in a fission yeast cell. Bottom: time series of the equatorial plane of dividing cells expressing GFP-Myo2. Dashed box: start of the contractile ring constriction. Number: time in minutes. (**E**) Time course of the fluorescence intensity of GFP-Myo2 at the equatorial plane after the start of contractile ring assembly (time zero). Cloud: standard deviations. Interval = 2 min. (**F**) Dot plot of the fluorescence intensity of GFP-Myo2 in a mature contractile ring just before it starts to constrict. **: *p* < 0.01. NS: not significant. Statistics were calculated using two-tailed Student’s *t* tests. All the data are pooled from at least two independent biological repeats. Scale bar = 5 µm.

**Figure 2 jof-10-00455-f002:**
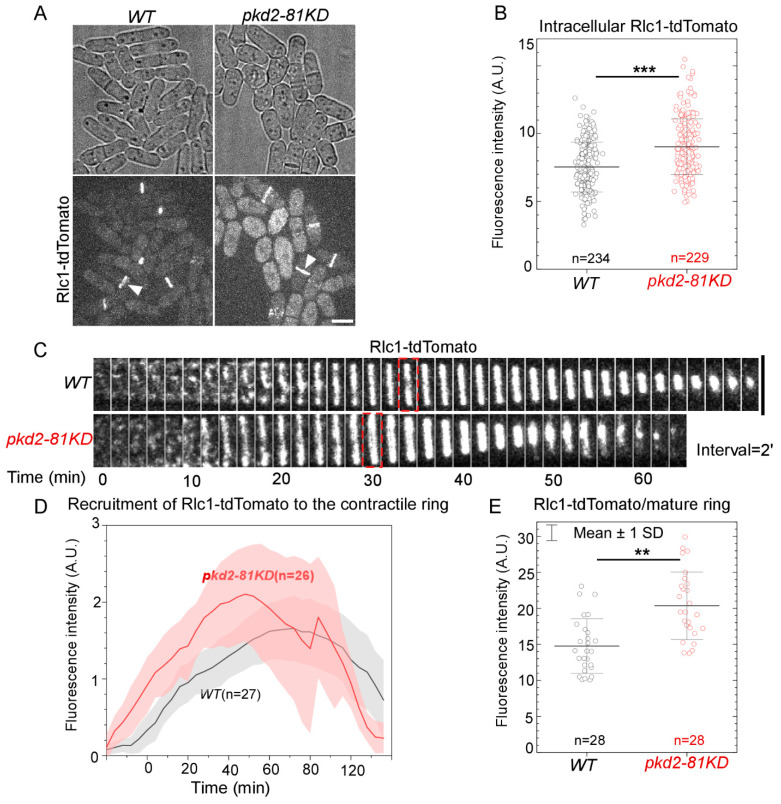
The *pkd2-81KD* mutant increased the recruitment of myosin II regulatory light chain Rlc1 to the contractile ring. (**A**) Micrographs of *wild type* and *pkd2-81KD* cells expressing Rlc1-tdTomato. White arrowhead: the contractile ring. (**B**) Dot plot of the intracellular fluorescence intensity of Rlc1-tdTomato in a cell. (**C**) Time series of the equatorial plane of dividing cells expressing Rlc1-tdTomato. Dashed box: start of the contractile ring constriction. Number indicates time in minutes. (**D**) Time course of the fluorescence intensity of Rlc1-tdTomato in the equatorial plane. Cloud: standard deviations. (**E**) Dot plot of the fluorescence intensity of Rlc1-tdTomato in a mature contractile ring. **: *p* < 0.01. ***: *p* < 0.001. Statistics were calculated using two-tailed Student’s *t*-tests. All the data are pooled from at least two independent biological repeats. Scale bar = 5 µm.

**Figure 3 jof-10-00455-f003:**
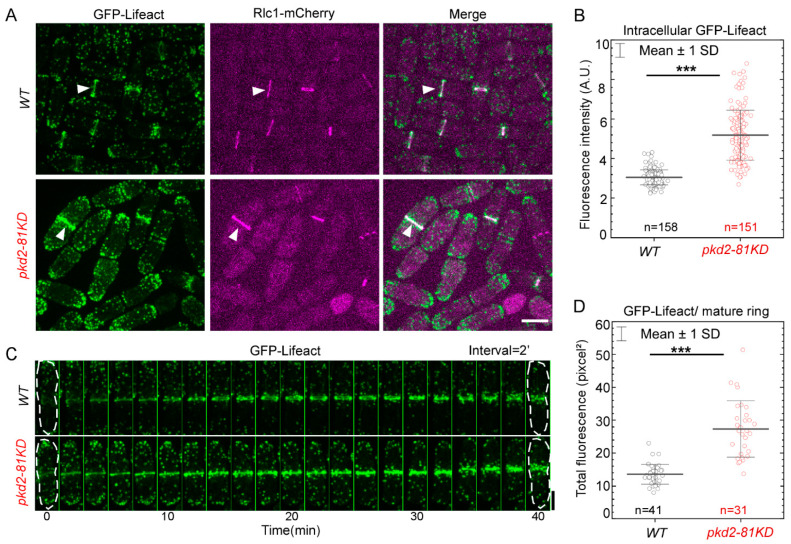
The *pkd2*-*81KD* mutation increased the assembly of actin filaments in the contractile ring. (**A**) Micrographs of the *wild type* (top) and *pkd2-81KD* (bottom) cells co-expressing GFP-Lifeact (green) and Rlc1-mCherry (magenta). White arrowhead: the contractile ring. (**B**) Dot plot of the average intracellular fluorescence intensity of GFP-Lifeact in a cell. (**C**) Time series of a *wild type* and a *pkd2-81KD* cell co-expressing GFP-Lifeact (green) and Rlc1-mCherry. Number: Time in minutes from the start of the contractile ring assembly based on the appearance of cytokinetic nodes marked by Rlc1-mCherry. (**D**) Dot plot of the total fluorescence of GFP-Lifeact in a mature contractile ring. ***: *p* < 0.001. Statistics were calculated using two-tailed Student’s *t* tests. All the data are pooled from three independent biological repeats. Bar = 5 µm.

**Figure 4 jof-10-00455-f004:**
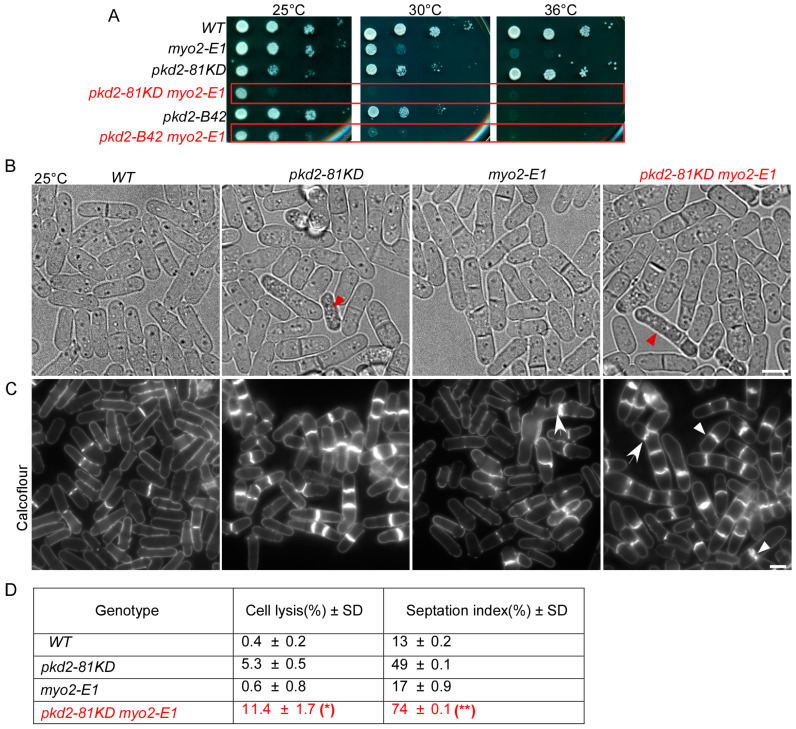
Negative genetic interactions between *pkd2* and *myo2* mutants. (**A**) Ten-fold dilution series of yeast cells at the indicated temperatures. (**B**) Bright-field micrographs of live cells at 25 °C. Red arrowhead: lysed cells. (**C**) Micrographs of calcofluor-stained fixed cells at 25 °C. Arrow: abnormal septum. Arrowhead: thick septum (n > 500). (**D**) A table summarizing the morphological defects of the *pkd2-81KD myo2-E1* mutant cells. All the data are pooled from at least two independent biological repeats. *: *p* < 0.05; **: *p* < 0.01. Statistics were calculated using two-tailed Student’s *t* tests. Scale bar = 5 µm.

**Figure 5 jof-10-00455-f005:**
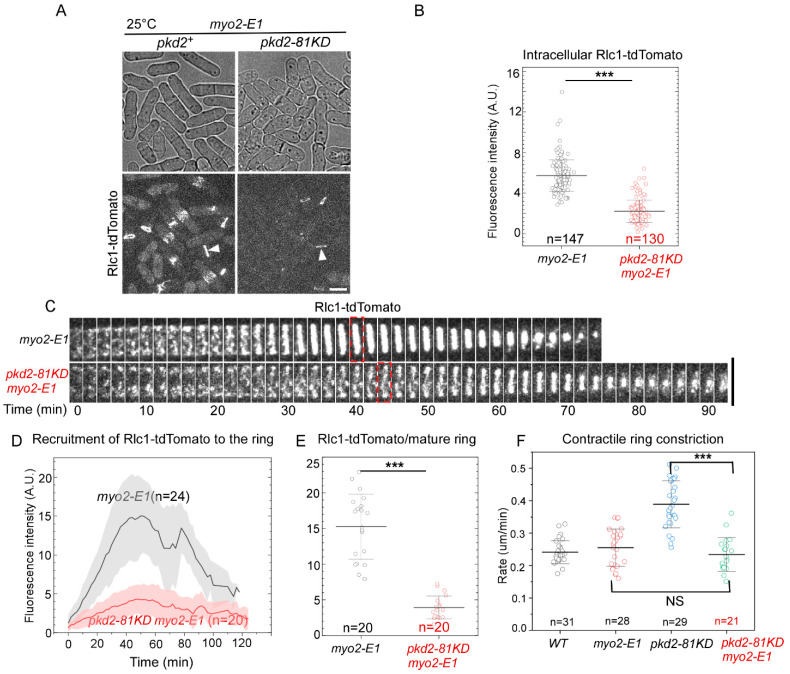
The *pkd2-81KD myo2-E1* mutant failed to assemble a complete contractile ring at the permissive temperature. (**A**) Micrographs of live cells expressing Rlc1-tdTomato. Arrowhead: the contractile ring. (**B**) Dot plot of average intracellular fluorescence intensity of Rlc1-tdTomato in a cell. (**C**) Time-lapse series of the equatorial plane of a *myo2-E1* and a *pkd2-81KD myo2-E1* cell expressing Rlc1-tdTomato. Dashed box: start of contractile ring constriction. Number: time in minutes. (**D**) Time course of the fluorescence intensity of Rlc1-tdTomato at the equatorial plane. Interval = 2 min. (**E**) Dot plot of average fluorescence intensity of Rlc1-tdTomato in a mature contractile ring before it starts to constrict. (**F**) Dot plot of the rate of the contractile ring constriction. All the experiments were carried out at the permissive temperature of 25 °C. ***: *p* < 0.001. NS: not significant. Statistics were calculated using two-tailed Student’s *t* tests. All the data are pooled from at least two independent biological repeats. Scale bar = 5 µm.

**Table 1 jof-10-00455-t001:** List of yeast strains used in this study.

Name	Genotype	Source
FY527	*h- leu1-32 ura4-D18 his3-D1 ade6-M216*	Lab stock
JW766	*h+ kanMX6-Pmyo2-GFP-myo2 ade6-M210 leu1-32 ura4-D18*	Lab stock
JW1341	*h- rlc1-tdTomato-natMX6 ade6-M210 leu1-32 ura4-D18*	Lab stock
QC-Y799	*h+ myo2-E1 rlc1-tdTomato-NatMX6 sad1-mGFP-KanMX6 ura4-D18 leu1-32 ade6-M?*	This study
QC-Y813	*h- kanMX6-81xnmt1-pkd2 rlc1-tdTomato-NatMX6 ura4-D18 leu1-32 ade6-M?*	Lab stock
QC-Y817	*h+ kanMX6-81xnmt1-pkd2 leu1-32 ura4-D18 his3-D1 ade6-M210*	Lab stock
QC-Y839	*h- myo2-E1 ura4-D?*	Lab stock
QC-Y840	*h+ myo2-E1 kanMX6-81xnmt1-pkd2 leu1-32 ura4-D18 his3-D1 ade6-M210*	This study
QC-Y1032	*h+ pkd2::pkd2-B42-ura4+-hist5+ leu1-32 ade6-M?*	Lab stock
QC-Y1033	*h? myo2-E1 kanMX6-81xnmt1-pkd2 rlc1-tdTomato-natMX6 ura4-D18 his3-D1 ade6-M210*	This study
QC-Y1112	*h? kanMX6-Pmyo2-GFP-myo2 ade6-M210 leu1-32 ura4-D18 kanMX6-81xnmt1-pkd2*	This study
QC-Y1752	*h? pkd2::pkd2-B42-ura4+-his5+ leu1-32 myo2-E1 ade6-M?*	This study
QC-Y1809	*h? rlc1-mCherry-natMX6 ade6-M210 leu1-32 ura4-D18 kanMX6-81xnmt1-pkd2 leu2::Kan-Padf1-GFP-Lifeact*	This study
QC-Y1810	*h? rlc1-mCherry-natMX6 ade6-M210 leu1-32 ura4-D18 leu2::Kan-Padf1-GFP-Lifeact*	This study

## Data Availability

The raw data supporting the conclusions of this article will be made available by the authors on request.

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
