# Peer review of "The Mechanosensitive Pkd2 Channel Modulates the Recruitment of Myosin II and Actin to the Cytokinetic Contractile Ring"

_jof, 2024, doi:10.3390/jof10070455_

Round 1
Reviewer 1 Report
Comments and Suggestions for Authors
Fission yeast is a widely used system model to study cytokinesis. In this study, Chowdury investigate the role of a mechanosensitive channel, Pkd2, in the assembly of contractile ring during cytokinesis. The authors take advantage of a hypomorphic mutant pkd2-81KD to assess the timing of contractile ring maturation apart from quantify several components of this dynamic structure. With the obtained data, the authors conclude that Pkd2 is required to modulate myosin II and actin filaments in the cytokinetic contractile ring. Most of the data presented is quite convincing. However, I found some issues that require clarification as follow:
Fig. 1A, E:
The authors state that GFP-Myo2 in whole cell is not affected in the pkd2 mutant. However, the picture presented in A suggest that GFP-Myo2 is indeed increased when compared to the wild-type. In any case, a Western blot for Myo2 would clarify this issue.
Fig. 3:
The title of the figure legend is wrong as it states a decrease of actin molecules in the pkd2-B42 mutant. However, the authors employed another mutant and actin seems to be increased.
Fig. 3C:
The authors state in line 170 “…actin molecules in the contractile ring of pkd2-81KD cells increased by 12% …” However, the statistics in the panel show that this is non-significant.
Fig. 4C legend:
Please indicate that numbers represent septation index calculations.
Fig. 4A:
The authors could also use minimal media lacking thiamine as control. I expect no phenotype displayed by pkd2-81KD.
Lines 97-103:
Please state the temperature used for cell growth.
Comments on the Quality of English Language_ Italicize QC-Y799 strain genotype.
_ Line 111: Please insert µ Greek character.
_ Line 283: “phosphorylate” must read “phosphorylates”.
_ Line 286: “is not essential the..” must read “is not essential for the..”.
Author Response
Reviewer 1 Comments and Suggestions for Authors
Fission yeast is a widely used system model to study cytokinesis. In this study, Chowdhury investigate the role of a mechanosensitive channel, Pkd2, in the assembly of contractile ring during cytokinesis. The authors take advantage of a hypomorphic mutant pkd2-81KD to assess the timing of contractile ring maturation apart from quantify several components of this dynamic structure. With the obtained data, the authors conclude that Pkd2 is required to modulate myosin II and actin filaments in the cytokinetic contractile ring. Most of the data presented is quite convincing. However, I found some issues that require clarification as follow:
Response: We would like to thank this reviewer for both careful reading of our manuscript and constructive critiques. We have now addressed all of the reviewer’s concerns through either experiments or textual revision.
Fig. 1A, E: The authors state that GFP-Myo2 in whole cell is not affected in the pkd2 mutant. However, the picture presented in A suggest that GFP-Myo2 is indeed increased when compared to the wild-type. In any case, a Western blot for Myo2 would clarify this issue.
Response: Thanks for pointing out this mistake in our original manuscript! We inadvertently displayed the two micrographs in Fig. 1A with different gray scales. Now we corrected this in the revised Fig. 1A. In it, the intracellular fluorescence of GFP-Myo2 appeared to be similar between the wild-type and the pkd2 mutant cells.
In addition, based on the reviewer’s suggestion, we also carried out anti-GFP immunoblots of the lysates of wild-type and pkd2-81kd cells expressing GFP-Myo2. We found similar expression levels of GFP-Myo2 in these two strains (Fig. 1C) (Line162 -163).
Fig. 3: The title of the figure legend is wrong as it states a decrease of actin molecules in the pkd2-B42 mutant. However, the authors employed another mutant and actin seems to be increased.
Response: That is a misunderstanding. Both Fig. 3C and 3D employed the same mutant pkd2-81KD. The only difference between these two panels is that we quantified the fluorescence of GFP-Lifeact in the contractile ring in C and in the whole cell in D. To avoid any confusion, we have revised the title of Fig. 3 as the following, “The pkd2-81KD mutant increased the assembly of actin filaments in the contractile ring” (Line 359-360).
Fig. 3C: The authors state in line 170 “…actin molecules in the contractile ring of pkd2-81KD cells increased by 12% …” However, the statistics in the panel show that this is non-significant.
Response: Thank you for pointing this out! We had made a mistake in our initial measurements of actin in the contractile ring. Further investigation in the process of this revision identified the error incurred in our initial data analysis. After correcting this, we now found the fluorescence of GFP-Lifeact in the contractile ring of pkd2-81KD was about 21% higher than the wild type cells(P<0.001) (Fig. 3D), similar to myosin II. We concluded that the number of actin molecules in the contractile ring of the pkd2 mutant cells increases (Line 202-204).
Fig. 4C legend: Please indicate that numbers represent septation index calculations.
Response: Thanks for pointing this out! For better representation, instead of numbers in the micrographs, we now added a new table summarizing the percentages of lysed cell and septated cells separately (Fig. 4D).
Fig. 4A: The authors could also use minimal media lacking thiamine as control. I expect no phenotype displayed by pkd2-81KD.
Response: Thanks for the great suggestion! Unfortunately, the pkd2 mutant did not grow particularly well, likely due to their inability to regulate the turgor pressure, in the minimal media. The osmolarity of EMM is about twice that of YE (our unpublished data). This prevented us from carrying out this suggested experiment. Nevertheless, the negative genetic interaction between another pkd2 mutant pkd2-B42 and myo2-E1 (Fig. 4A) provided confirmation to the synergistic relationship between pkd2 and myo2.
Lines 97-103: Please state the temperature used for cell growth.
Response: We have added the temperature as suggested by the reviewer.
Comments on the Quality of English Language
_ Italicize QC-Y799 strain genotype.
_ Line 111: Please insert µ Greek character.
_ Line 283: “phosphorylate” must read “phosphorylates”.
_ Line 286: “is not essential the..” must read “is not essential for the..”.
Response: Thanks for careful reading of our manuscript! We have corrected these typos as suggested by the reviewer.
Reviewer 2 Report
Comments and Suggestions for Authors
Comments on the quality of writing are included with the "Comments and suggestions for authors".
Author Response
Reviewer 2:
In this work, Chowdhury et al. investigate the impact of the ion channel Pkd2 on cytokinesis. To achieve this goal, they measure the fluorescence intensity of labelled Myo2, Rlc1 and of a Lifeact actin probe in the contractile ring. They also determine the genetic interaction between pkd2-81KD and myo2-e1. The results shown support a role for Pkd2 in the recruitment of Myo2 and Rlc1 to the contractile ring but not of actin filaments. In addition, they notice a genetic interaction between pkd2-81KD and myo2-e1 in the assembly and constriction of the contractile ring. Overall, the information contained in this work is valuable to the field and enhance our understanding of the function of Pkd2 during cytokinesis.
Response: We would like to thank the reviewer for both careful consideration of our manuscript and for supporting its publication. We have now addressed all the critical issues raised by the reviewer in this revised manuscript. A point-by-point rebuttal is below.
Major concerns
Critical issues with the manuscript
- Title and many lines including 22, 84, 139, 152, 251, 266, 303 and 316: No experiments in this work support that Pkd2 promotes or modulates the “assembly” of Myo2 molecules. The experiments performed support that Pkd2 modulates or control the “recruitment” of Myo2 to the contractile ring during constriction. The title needs to be changes and the text amended accordingly. For example, the following title would be more appropriate “The mechanosensitive channel Pkd2 modulates the recruitment of Myo2 to the cytokinetic contractile ring.”
Response: Thanks for raising this critical point! Based on the reviewer’s suggestion, we have removed most references to “assembly” from the manuscript. For example, we have revised our title as “The mechanosensitive Pkd2 channel modulates the recruitment of myosin II and actin to the cytokinetic contractile ring”.
- Title and many other lines across the manuscript: The data in Figure 3 shows no significant difference in the amount of actin filaments in the contractile ring of wild type and pkd2-81KD cells. Therefore, the current title must be rephrased to eliminate the statement that Pkd2 modulates actin filaments in the contractile ring as it is unsupported by the data. Please see the title suggested in point 1 for a more appropriate option.
Response: Thanks for pointing this out! We had made a mistake in our initial measurement of actin molecules in the contractile ring. Further investigation in the process of this revision identified the error originated from our initial data analysis. After correcting this we now found the fluorescence of GFP-Lifeact in the contractile ring of pkd2-81KD was about 21% higher than wild type cells (P<0.001) (Fig. 3D). We concluded that the number of actin molecules in the contractile ring increases in the pkd2 mutant cells. (Line 202-204).
Lack of background on Pkd2 in the introduction.
- The introduction lacks sufficient background on Pkd2 in fission yeast cytokinesis. The authors refer to their previous publications across the text but a section on their findings on Pkd2 is important and would support the significance of the findings of this work.
Response: Thanks for this great suggestion! We have now added a paragraph that detailed our works on the fission yeast Pkd2 and its role in cytokinesis as a mechanosensitive calcium influx channel in the introduction (Line 46-65).
- The paragraph on polycystins PC-1 and PC-2 appears to have no purpose in the manuscript. The information contained in that paragraph is not used in the rest of the manuscript and is therefore unnecessary to the results or the discussion. Either be more specific with the need for this information or remove this paragraph.
Response: Based on the reviewer’s suggestion, we have removed irrelevant information about PC1 and PC2 from the introduction of the revised manuscript.
Lack of interpretation in the discussion.
- Based on the text, the authors expected Pkd2 to promote Rlc1 recruitment to the ring as Rlc1 is a Ca++ binding protein. However, the results show the opposite outcome from that expected as the depletion of Pkd2 resulted in the increase of Rlc1 both in the contractile ring and in the cytoplasm. Please clarify this conundrum in the results and address this interesting finding in the discussion.
Response: The reviewer raised a great point in regard to the recruitment of Rlc1. We added the following in the discussion (Line 292-294). “Alternatively, Pkd2 could indirectly regulate transcription through calcineurin. The Ca2+/calmodulin-dependent phosphatase Ppb1 promotes the nuclear translocation and activation of the transcription activator Prz1 (Hirayama et al., 2003) .”
- *Critical* and linked to concern 2 above. The data in Figure 3C show no significant difference in the fluorescence intensity of the actin probe in wild type versus pkd2-81KD. Yet, the text mentions a “12% increase” in the amount of actin, yet the data show no significant change in the amount of actin filament in the ring. The sentence should mention that there is “no difference in the amount of actin filaments” between wild type and pkd2-81KD mutant.
Response: Thanks for pointing out this critical point! We had made a mistake in our initial quantification of actin molecules in the contractile ring. Further investigation in the process of this revision identified the error made in our initial data analysis. After correcting this, we now found the fluorescence of GFP-Lifeact in the contractile ring of pkd2-81KD was about 21% higher than wild type cells (P<0.001) (Fig. 3D). We concluded that the number of actin molecules in the contractile ring increases in the pkd2 mutant cells. (Line no:202-204).
- Stark et al. 2010 demonstrated that increasing the amount Myo2 in the contractile ring resulted in a slower constriction rate. In your work, you suggest that the faster constriction rate of rings in pkd2 mutant cells is caused by a 20% increase in Myo2, which opposes the conclusions from Stark et al. Provide an explanation for this difference in results in the discussion.
Response: Thanks for pointing us to this highly relevant paper by the group of Matt Lord! Based on the reviewer’s suggestion, we have now added this in the discussion (Line 258-260 ) that “Although the increase of the heavy chain by itself has little effect on the constriction of the contractile ring (Stark et al., 2010), the increased densities of both Myo2 and the regulatory light chain Rlc1 as well as actin filaments may.”
- In Malla et al. 2022, contractile rings in myo2-e1 cells show a decrease in the total amount of actin filaments in the ring. In this work, the 20% increase in Myo2 is not accompanied by a significant increase in actin filaments in the contractile ring. Please provide an interpretation of this interesting findings in the discussion.
Response: The reviewer raised an interesting point! Our newly analyzed data is indeed consistent with the reviewer’s prediction. Both Myo2 and actin increased by a similar margin of 20% in the contractile ring of the pkd2 mutant cells. To highlight this point, we added the following in the discussion (Line 256-257) “The increase of Myo2 in the ring of the pkd2 mutant cells is proportional to the increase of actin filaments in the contractile ring.”
- A potential reason for the increased in the cytoplasmic levels of actin in pkd2 cells is an impact on the expression levels of the Lifeact probe. Since this hypothesis has not been tested, it should be reported as a potential interpretation.
Response: Based on the reviewer’s suggestion, we have now added the following to report the potential increased expression of GFP-Lifeact in the discussion (Line 285-287 ) “Specifically, the significant increase of intracellular concentration of GFP-Lifeact in the pkd2 mutant cells suggests that the expression of this actin probe may be affected by the pkd2 mutation.”
Major issues with the result section
- The annotation “pkd2 myo2” to refer to the double mutant strain studied in this work is wrong and can be found throughout most of the text. It must be replaced with the complete annotation of the alleles pkd2-81KD myo2-e1.
Response: Thanks for pointing this mistake out! We have corrected it throughout the text.
- Lines 41-42: What do you mean by a role in disassembly?
Response: We meant that Myo2 promotes the disassembly of actin filaments from the contractile ring. We have now clarified this statement as the following “… disassembly of actin filaments…” (Line 38)
- Images that are magenta on black in Figures 2A, 2B, 3A, 5A, 5B are too dark to show the phenotypes described. Figure 5B barely shows anything for the mutant. Please use white on black as in Figure1 unless a figure panel shows the merge of two colors.
Response: Based on the reviewer’s suggestion, we replaced all the magenta-colored micrographs in Fig. 2 and 5 with gray colored ones.
- Lines 241-242 and 250-251. The authors conclude that Pkd2 and Myo2 regulate the contractile ring assembly and the recruitment of Rlc1 to the ring though “complementary pathways”. There is no reasoning provided why they act in complementary pathway. Explain how your data supports genetic complementarity to support this interpretation. Such explanation may be best suited for the discussion. In fact, the myo2-e1 mutation rescues the constriction rate phenotype of pkd2-81KD.
Response: Based on the reviewer’s suggestion, we have now removed the wording to “Complementary Pathways” from this revised manuscript. In its place, we discussed how Pkd2 and Myo2 may both contribute to the stability of the contractile ring.
- Lines 244-245 and 247. It is stated that the authors measured the “number of Rlc1-tdTomato molecules” yet only fluorescence intensities are shown in Figures 5D to F. Please fix the text accordingly.
Response: Thanks for this great point! We have now instead stated that the fluorescence intensities of tdTomato in the contractile ring of the pkd2 mutant cells are higher than the wild type.
- Line 266 and others. This work shows no evidence that Pkd2 promotes the “assembly” of Myo2. This work instead supports that Pkd2 regulates the “recruitment” of Myo2 to the contractile ring. This comment is related to comment 1.
Response: Thanks for raising this point! We have now instead concluded that Pkd2 modulates the recruitment of Myo2 to the contractile ring throughout the text.
- Line 275. “This phenotype is in line with our new finding that the contractile ring of the mutant cells contains 20% more Myo2, Rlc1 and actin than the wild type ones.” This sentence is wrong. There is no significant difference in the amount of actin filaments in the ring in the mutant versus wild type as shown in Figure 3C. This comment is related to comment 2.
Response: Thanks for raising this point! We made a mistake in our initial data analysis of the Lifeact-GFP in the contractile ring. In this revised manuscript, after we corrected the mistake, we now found the fluorescence of GFP-Lifeact in the contractile ring of pkd2-81KD was about 21% higher than the wild type cells (P<0.001). We concluded that the number of actin molecules in the contractile ring increases by 21% in the pkd2 mutant cells (Fig. 3D) (Line 203-204).
Minor concerns
There are many typos, writing and figure annotation mistakes. I caught the following but please carefully proofread your manuscript.
Line 61: Evolutionally. Do you mean “evolutionarily”?
Line 95: “Spotting”. This appears to be a typo. Do you mean “being spotted”?
Table of strains: QC-Y799. Should be in italic font.
Line 197: The title of the figure legend mentions that the allele used was pkd2-B42. Yet all 4 panels refer to pkd2-81KD. Please fix this mistake.
Line 228. “Arrow: lysed cells”. The figure panel shows an arrowhead. Please fix the legend.
Line 229. “Arrow”. The arrows in the figure look like arrowhead causing confusion with what the authors are trying to identify. Fix the arrows so the “tails” are more apparent.
Line 230. “Arrowheads: curved septum”. The arrowheads highlight thick septa perhaps, but they don’t look curved, at least not at this magnification.
Lines 281-282: The following sentence needs fixing: “The modulating effect of Pkd2 in the assembly of contractile ring most likely is rooted its activity as a Ca2+-permeable ion channel.”
Response: Thanks for pointing it out. We have corrected the sentence (Line 265-266).
Lines 286-287: The following sentence needs fixing: “Although such phosphorylation of Rlc1 is not essential the temporal regulation of cytokinesis”
Line 299: This sentence appears to have been forgotten in the text: “Ca2+-sensitive phosphotase calcineurin.”
Lines 289 and 299: Phosphotase is a mistake. It’s phosphatase.
Line 310-311: Do you mean “compromised” in the following sentence? “In contrast, Myo2 promotes the contractile ring stability 310 through its motor activity, which is comprised in the myo2-E1 mutant”.
Reference 4 appears to have been affected by formatting
Response: We thank the reviewer for careful reading of our manuscript! We have fixed all these minor issues pointed out by the reviewer.
Reviewer 3 Report
Comments and Suggestions for Authors
In my opinion, the experimental and analytical methodology are suitable and reliable and the manuscript can be accepted after a thorough typographical review. These errors are marked in the attached revised text.

Author Response
Reviewer 3:
Comments and Suggestions for Authors
In my opinion, the experimental and analytical methodology are suitable and reliable and the manuscript can be accepted after a thorough typographical review. These errors are marked in the attached revised text.
Response: We thank the reviewer for careful consideration of our manuscript and for supporting its publication. We have corrected all the typos pointed out by the reviewer.
Major comments:
Line 287,288: This cannot be concluded exactly from the reading of Prieto et al.2023. Actually, Rlc1 phosphorylation at S35 becomes critical for adequate cytokinesis when actin filament availability is compromised,
Response: Thanks for pointing to these two critical studies about the phosphorylation of Rlc1 by Preito-Ruiz et al. in 2023. To incorporate their findings, we added the following to the discussion, “Alternatively, the phosphorylation of Rlc1 may be regulated through a balance of phosphorylation by Pak1/2 and dephosphorylation by the calcium/calmodulin-activated phosphatase calcineurin Ppb1.” (Line 278-280).
We thank the reviewer for careful reading of our manuscript! We have fixed all these minor issues pointed out by the reviewer. We have corrected the marked issues in the manuscript file which we have attached here.

Round 2
Reviewer 1 Report
I am satisfied with the authors' response.
I am satisfied with authors' response.
Author Response
We thank the reviewer for approving the revised manuscript after the first round of revision.
Reviewer 2 Report
The improvements to the manuscript are greatly appreciated. The entire text reads better with the modifications made by the authors, and the significance of the results are more apparent. The following concerns remain to be addressed before the work is suitable for publication.
Major concerns:
Line 242-249: I’m still not convinced that there’s an increase in actin filaments in the ring of pkd2-81KD cells. Because the expression of the GFP-Lifeact probe increases ~90% in pdk2-81KD cells, one cannot tell how the labeling to the ring compares with wild-type. A much higher cytoplasmic concentration of the GFP-Lifeact will result in an increased decoration of the actin filaments in the ring even without a change in the amount of actin filaments in the ring. In addition, such higher expression levels may affect the stability of actin filaments in the cells. Although the authors may prefer the interpretation that actin filaments are increased in the rings of pkd2-81KD cells, they must include that it is possible that the increased fluorescence in the ring is due to the increased expression of the GFP-Lifeact probe in the cell.
Lines 296-298: From the way the text is written, it appears that the authors suggest that there are fewer cells with rings in the population. That seems unlikely as they previously established that the septation index in cells is increased. Because the following sentence, which begins at line 298, refers to the fluorescence intensity of the Rlc1-tdTomato probe, they meant that the fluorescence intensity of Rlc1-tdTomato is decreased. Please revised this section and clarify if you found that fewer cells had rings in the population or if the rings showed a lower fluorescence intensity of Rlc1-tdTomato.
Lines 325-328: The reference from Stark et al. is mis-reported. Constriction rate was measured to be significantly reduced in cells expressing twice the amount of Myo2. You can write “Although doubling Myo2 causes a decrease in the constriction rate, other ring components were not measured… More work is needed to determine how the concentration of Myo2 in the ring affects constriction rate.”.
Line 374: Balasubramanian et al. did not show evidence that Myo2-E1 has a compromised motor activity. The best knowledge we have of this allele is from Lord and Pollard, 2004 who showed that purified Myo2-E1 has a defect in actin filament binding. Because Myo2-E1 cannot bind actin filaments in vivo, motor activity cannot be measured. You can change “motor activity” for “actin binding” in the sentence.
Minor concerns:
Line 73: Instead of using the term “inducible”, I suggest using the term “repressible”, which is acceptable for the nmt promoter. It is easier to understand the method of the experiment if the promoter is termed “repressible”.
Line 268: The term “morphogenesis” is wrong. I think the authors meant “morphology”.
Line 351: Please revise the sentence that starts with “In addition to direct modulation of the recruitment of myosin and actin…” Looks like a word is missing.
The improvements to the manuscript are greatly appreciated. The entire text reads better with the modifications made by the authors, and the significance of the results are more apparent. The following concerns remain to be addressed before the work is suitable for publication.
Major concerns:
Line 242-249: I’m still not convinced that there’s an increase in actin filaments in the ring of pkd2-81KD cells. Because the expression of the GFP-Lifeact probe increases ~90% in pdk2-81KD cells, one cannot tell how the labeling to the ring compares with wild-type. A much higher cytoplasmic concentration of the GFP-Lifeact will result in an increased decoration of the actin filaments in the ring even without a change in the amount of actin filaments in the ring. In addition, such higher expression levels may affect the stability of actin filaments in the cells. Although the authors may prefer the interpretation that actin filaments are increased in the rings of pkd2-81KD cells, they must include that it is possible that the increased fluorescence in the ring is due to the increased expression of the GFP-Lifeact probe in the cell.
Lines 296-298: From the way the text is written, it appears that the authors suggest that there are fewer cells with rings in the population. That seems unlikely as they previously established that the septation index in cells is increased. Because the following sentence, which begins at line 298, refers to the fluorescence intensity of the Rlc1-tdTomato probe, they meant that the fluorescence intensity of Rlc1-tdTomato is decreased. Please revised this section and clarify if you found that fewer cells had rings in the population or if the rings showed a lower fluorescence intensity of Rlc1-tdTomato.
Lines 325-328: The reference from Stark et al. is mis-reported. Constriction rate was measured to be significantly reduced in cells expressing twice the amount of Myo2. You can write “Although doubling Myo2 causes a decrease in the constriction rate, other ring components were not measured… More work is needed to determine how the concentration of Myo2 in the ring affects constriction rate.”.
Line 374: Balasubramanian et al. did not show evidence that Myo2-E1 has a compromised motor activity. The best knowledge we have of this allele is from Lord and Pollard, 2004 who showed that purified Myo2-E1 has a defect in actin filament binding. Because Myo2-E1 cannot bind actin filaments in vivo, motor activity cannot be measured. You can change “motor activity” for “actin binding” in the sentence.
Minor concerns:
Line 73: Instead of using the term “inducible”, I suggest using the term “repressible”, which is acceptable for the nmt promoter. It is easier to understand the method of the experiment if the promoter is termed “repressible”.
Line 268: The term “morphogenesis” is wrong. I think the authors meant “morphology”.
Line 351: Please revise the sentence that starts with “In addition to direct modulation of the recruitment of myosin and actin…” Looks like a word is missing.
Author Response
Reviewer 2 Comments and Suggestions for Authors
The improvements to the manuscript are greatly appreciated. The entire text reads better with the modifications made by the authors, and the significance of the results are more apparent. The following concerns remain to be addressed before the work is suitable for publication.
Response: We thank the reviewer for the positive review of our revised manuscript. We have now addressed all of the reviewer’s concerns through textual revisions. We hope that the reviewer finds our manuscript is now ready for publication.
Major concerns:
Line 242-249: I’m still not convinced that there’s an increase in actin filaments in the ring of pkd2-81KD cells. Because the expression of the GFP-Lifeact probe increases ~90% in pdk2-81KD cells, one cannot tell how the labeling to the ring compares with wild-type. A much higher cytoplasmic concentration of the GFP-Lifeact will result in an increased decoration of the actin filaments in the ring even without a change in the amount of actin filaments in the ring. In addition, such higher expression levels may affect the stability of actin filaments in the cells. Although the authors may prefer the interpretation that actin filaments are increased in the rings of pkd2-81KD cells, they must include that it is possible that the increased fluorescence in the ring is due to the increased expression of the GFP-Lifeact probe in the cell.
Response: Thanks for raising this point! As suggested by the reviewer, we added the following in the discussion section “The elevated expression of this fluorescence actin probe in the pkd2 mutant cells may partially contribute to the increased fluorescence of GFP-Lifeact in the contractile ring.” (Line 288-289). We think that both increased assembly of actin filaments and elevated expression of GFP-Lifeact contribute to the increased fluorescence of this reporter in the ring.
Lines 296-298: From the way the text is written, it appears that the authors suggest that there are fewer cells with rings in the population. That seems unlikely as they previously established that the septation index in cells is increased. Because the following sentence, which begins at line 298, refers to the fluorescence intensity of the Rlc1-tdTomato probe, they meant that the fluorescence intensity of Rlc1-tdTomato is decreased. Please revised this section and clarify if you found that fewer cells had rings in the population or if the rings showed a lower fluorescence intensity of Rlc1-tdTomato.
Response: The reviewer has a good point. We have removed the description that there are less contractile rings among the pkd2-81KD myo2-E1 mutant cells. Instead, we stated the following, “the fluorescent intensities of Rlc1-tdtomato in the contractile rings of pkd2-81KD myo2-E1 cells appeared to be far lower than that of myo2-E1 cells” (Line 229-231).
Lines 325-328: The reference from Stark et al. is mis-reported. Constriction rate was measured to be significantly reduced in cells expressing twice the amount of Myo2. You can write “Although doubling Myo2 causes a decrease in the constriction rate, other ring components were not measured… More work is needed to determine how the concentration of Myo2 in the ring affects constriction rate.”.
Response: Thanks for pointing it out! We have added the suggested text to the revised manuscript based upon the reviewer’s suggestion (Line 258-260).
Line 374: Balasubramanian et al. did not show evidence that Myo2-E1 has compromised motor activity. The best knowledge we have of this allele is from Lord and Pollard, 2004 who showed that purified Myo2-E1 has a defect in actin filament binding. Because Myo2-E1 cannot bind actin filaments in vivo, motor activity cannot be measured. You can change “motor activity” for “actin binding” in the sentence.
Response: Thanks for raising this point! Based on the reviewer’s suggestion, we have revised the text accordingly (Line 307).
Minor concerns:
Line 73: Instead of using the term “inducible”, I suggest using the term “repressible”, which is acceptable for the nmt promoter. It is easier to understand the method of the experiment if the promoter is termed “repressible”.
Response: Based on the reviewer’s suggestion, we instead used “repressible” to describe nmt1 promoter throughout.
Line 268: The term “morphogenesis” is wrong. I think the authors meant “morphology”.
Response: Based on the reviewer’s suggestion, we have corrected this in the text.
Line 351: Please revise the sentence that starts with “In addition to direct modulation of the recruitment of myosin and actin…” Looks like a word is missing.
Response: Thanks for pointing out. We have added the “In addition to direct modulation of the recruitment of myosin II and actin to the contractile ring…” (Line 283-284).
Round 3
Reviewer 2 Report
The revised manuscript is ready to be considered for publication.
There are no further comments from this reviewer.
Author Response
We have corrected all the concerns and highlighted them all in manuscript files.